



# Dissolved organic carbon vertical movement and carbon accumulation in West Siberian peatlands

Evgeny A. Zarov[1], Elena D. Lapshina[1], Iris Kuhlmann[2], Ernst-Detlef Schulze[2]

[1]UNESCO chair «Environmental dynamic and global climate changes», Yugra State University,
Khanty-Mansiysk, 628012, Russian Federation
[2]Max Planck Institute for Biogeochemistry, P.O. Box 100164, 07701 Jena, Germany

*Correspondence to*: Evgeny A. Zarov (zarov.evgen@yandex.ru)

**Abstract.** Dissolved organic carbon is an additional path of carbon cycle but there is a lack of information about its distribution in peatland and rates of downward movement. We dated seven
peat cores (separately the dissolved (DOC) and particulate (POC) organic carbon) from Mukhrino peatland (typical zonal oligotrophic bog) in western Siberia to assess the date distribution between those two peat fractions. Our results revealed that the DOC is younger than POC for the surface peatland layers (0-150 cm) and older for the deeper layers. The date differences increases with depth and reaches 2000-3000 years at the bottom layer (430-530 cm). In our hypothesis this date
discrepancy caused by more young DOC moving to the deeper and older peat layers. The estimated average value of DOC downward movement was $0.047\pm0.019$ cm yr$^{-1}$.

Th oldest dates found at the lake bottom and ancient riverbed were 10 053 and 10 989 cal yr BP correspondingly. For the whole period of peatland functioning the average peat accumulation rate was estimated as $0.067\pm0.018$ cm yr$^{-1}$ (0.013–0.332 cm yr$^{-1}$), the carbon accumulation rate was
20 estimated as $38.56\pm12.21$ g C m$^{-2}$ yr$^{-1}$ (28.46–57.91 g C m$^{-2}$ yr$^{-1}$).

## 1. Introduction

Peatlands unlike the most other ecosystems assimilate carbon and sequestrate it over thousands years as long as  net primarily production exceeds the rate of organic matter decomposition. It has
25 been estimated that peatlands occupy about 2.84% (4.23 million sq. km.) of global land area (Xu, 2018) but have accumulated disproportionally huge amount of world soil carbon (~30 %) (Yu, 2014).

West Siberia is the world's largest wetland where peatlands cover 50–75 % of whole area (Peregon, 2008) that makes this territory one of the most waterlogged place in the world. Peatlands
occupy here not only local relief depressions but also vast watershed areas and floodplains (Bleuten and Lapshina, 2001). The West Siberian peatlands are estimated to contain ~20 % of world peat

deposits with the highest coverage percent in the taiga zone (Kremenetski, 2003; Sheng, 2004) having a carbon stock of 70.2 Pg (up to ~26 % of all terrestrial carbon) (Smith, 2004).

Carbon sequestration by peatlands and losses through surface runoff and greenhouse gases emission

are strongly correlated to the climate conditions (Baird et al., 2013). Part of litter and peat convert to dissolved form (so-called dissolved organic carbon, DOC) when in contact with exo-enzymes of microorganizms and acidic peatland water (Aravena et al., 1993; Charman et al., 1994,1999; Chasar et al., 2000; Clymo and Bryant, 2008, Schulze et al., 2015; Bleuten et al., 2020).

The release of DOC to streams and rivers is a characteristic phenomenon of peatland landscapes

(Frey and Smith, 2005). DOC discharging into surface water from the upper peatland layers derived from recently fixed C  (~50 years) (Billett et al., 2007). However, there is also old DOC flowing vertically up and down in the peat profile and its proportion is still unclear. Expected global warming may cause elevated concentrations of DOC discharge to the streams and significantly increase DOC flux to the Arctic Ocean (Freeman et al., 2004).

The process of DOC downward movement may lead to the mixing of young and old C and therefore dating inversions. There are several well-known possible reasons causing such inversions, such as roots intrusion (Glaser et al. 2012), slipping from the neighboring peat (Jaworski and Niewiarowski, 2012), peat fires (Turetsky et al., 2011) and other profile disturbances (Kołaczek, et al., 2019) such as dry years, cryoturbation, periodical flooding or other factors (Väliranta, et al., 2014). Usually the

inversed dates are excluded from the age-depth model which may lead to the significant shift of the model shape and different interpretation scenarios (Lamentowicz et al., 2015; Kołaczek, et al., 2019). All mentioned reasons happen occasionally, while the DOC flux is expected to be permanent and causes an error of under- or overestimation of the sample date. In this study we focus on the effects of a slow vertical downward movement of water. Our data may help to increase accuracy of

[14]C dating and to give a possible explanation of outliers in the age-depth models.

In this research we aim: (1) to show differences between dating of dissolved and particle organic carbon; (2) to estimate a velocity of DOC downward movement; (3) to assess the peat and carbon accumulation rates.

Our *hypothesis* is that DOC flows down into the mineral base sediment has a younger age and thus

may cause inversion of dating of bulk samples.

## 2. Study site

### 2.1. Peatland location

Mukhrino peatland is located in the middle taiga zone (Gvozdetskii et al., 1973) of West Siberia near by the confluence of two huge rivers Irtysh and Ob' (60.889 N, 68.702 E). It occupies a local



watershed between two small rivers "Mukhrina" and "Bolshaya rechka" on the left terrace of Irtysh
River (Fig. 1. cores_location.png). This is a zone of oligotrophic (rain fed) raised *Sphagnum* bogs
(Ivanov and Novikov, 1976). The total Mukhrino peatland area is 65 km² which is operated by the
Mukhrino field station (UNESCO Chair of Environmental Dynamic and Global Climate Changes,
Yugra State University) and situated in 20 km south-west from Khanty-Mansiysk city.

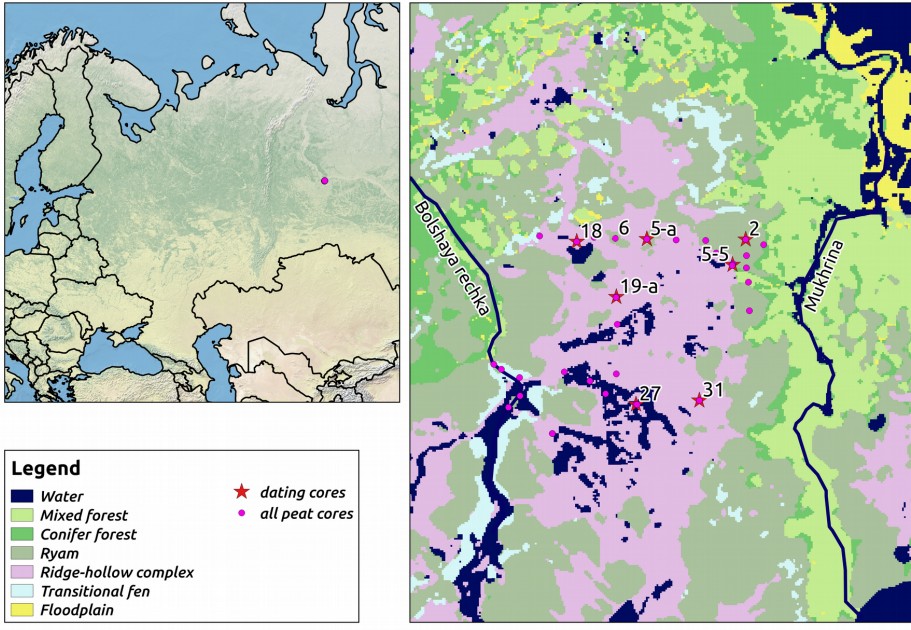

Figure 1: Location of the Mukhrino field station and all cores.

## 2.2. Climatic conditions

Climate of the region is moderately continental. According to the data of the station of the Russian
Weather Service at Khanty-Mansiysk the annual air temperature is –0.9° C (–1.0° C*[1]) with a cold

winter with the mean January temperature –20.0 C (–21.5° C*) and a short summer with the mean
July temperature 18.2 C (17.4° C*). Mean annual amount of precipitation is 533 mm (480 mm*)
with the most of this volume 314 mm (307 mm*) falling at the summer-autumn season (mostly in
August-September). The rest of precipitation is snow which appears at the end of October, starts
melting in the mid of April and completely disappears in May. The area is characterized by the

absence of permafrost, but spots of frozen soil on the depth 50–100 cm might be found till the end
of June.

---

[1]Values calculated on the base of measurements at the Mukhrino field station for the 2009-2020 years (Dyukarev et al.,
2021).





### 2.3. Vegetation

The peatland is surrounded by taiga forest dominated by *Pinus sibirica, Picea obovata, Abies sibirica* mixed with *Populus tremula, Betula pubescens*, occupying the mineral islands and up-land

slopes along the rivers and streams.

The main peatland vegetation unit types are:

Pine-dwarf shrub-peat moss community, so-called *"ryam"*, **typical ryam** is characterized by low Scotish pine leyer (*Pinus sylvestris* f. *litwiniwii*, 1.5–4 m height), well developed dwarf shrub layer (*Ledum palustre, Chamaedaphne calyculata*), and dominance of *Sphagnum fuscum* in moss layer

with minor admixture by *S. angustifolium* and *S. divinum*;

**Tall ryam** with similar to the typical ryam vegetation but with tall pine trees (*Pinus sylvestris* f. *uliginosa*, 6-10 m height) and *Sphagnum angustifolium* dominance in moss layer. Tall ryam occupies shallow peat deposits on the peatland outskirts.

**Ridge-hollow complex** featured by alternating typical ryam communities occupying elongated

across water flow ridges and waterlogged sedge-peat moss hollows (*Carex limosa, Scheuchzeria palustris, Eriophorum russeolum, Sphagnum balticum, S. majus, S. jensenii*);

Treeless **through flow fens and Sphagnum lawns** with hollow vegetation and rare scattered hummocks are developed within limited areas in the lower reaches of the peatland water catchment (Filippov and Lapshina, 2008).

Peatland has a dome shape with the difference in elevation between the edge and central parts in 1.2 meters (Bleuten et al., 2020). Central part is occupied by ridge-hollow complex; at the edge it becomes more inclined and drained letting the ryam and tall-ryam communities grow.

### 2.4. Peatland stratigraphy

Mukhrino peatland has been initiated as a minerotrophic fen with dominance of trees (birch, pine,

fir) and herbs (fern, horse-tail and tussock-forming sedges). Remains of these plants form the bottom layer of minerotrophic peat covering whole area of the peatland. Its thickness does not exceed 1.7 m and is 0.65 m in average. A thin layer (~0.5 m) of transitional peat formed by mesotrophic plant remains (*Scheuchzeria palustris*, sedges, dwarf shrubs and *Sphagnum* mosses) overlays the bottom peat layer. Oligotrophic peat forms the main upper part of peatland body. About

2/3 of peat deposit is composed of *Sphagnum* peat with the interlayer of thin cotton-grass-*Sphagnum* or sedge-*Scheuchzeria-Sphagnum* peat types, as formed by dynamic changes of the ridge-hollow complex.

The most abundant peat types are *Sphagnum fuscum*-peat (share in the structure of the peat deposit 22.5%), *Sphagnum* hollow peat (*S. balticum, S. papilosum* (12.0%)) and mixed *Sphagnum*

oligotrophic peat (*S . fuscum, S. angustifolium, S. divinum, S. papillosum, S. balticum* (5.7%)).



### 2.5. Peatland hydrology

Mukhrino peatland is fed by the rain and melt snow water. The highest water level is coincided with the snow melt at the end of April – beginning of May. Usually melt water is kept on the peatland surface in local depressions or blocked by ridges until the upper peat layer melt down. In that time
all water rapidly seeps into the peatland, dramatically increasing water level. The lowest water level is recorded at the end of summer time (usually in August) then water table rise up because of intense autumn precipitation and low air temperature decreasing an evapotranspiration. Discharge from the streams stops at the mid of October when water freezes. For more information see (Bleuten et al., 2020).

**3. Materials and methods**

### 3.1. Field sampling

In summer 2016, a total of 7 peat cores (Table 1) were extracted through the entire depth of peatland with a Russian corer (5.0 cm inside diameter, 0.5 m sampling step). Each half-meter part was moved to the plastic cassette, wrapped in stretch-film, transported to the laboratory (Yugra State
University) and subdivided into 10-cm subsamples for the further analysis. In the field 68 samples of 1 cm thick were cut from the bottom of each half-meter of 7 peat cores for the AMS radiocarbon analysis. With minimum contact to the environmental to avoid contamination it has been moved to the plastic zip-bags, labeled and sent to the Max-Plank Institute of Biogeochemistry in Jena (Germany).

Table 1. Peat sampling point description

| N | Core title | Habitat description | Water table, cm | Peat depth, cm |
|---|---|---|---|---|
| 1. | Core 2 | Typical ryam – dry peatland merge; covered by pine trees (up to 3 m high), *Ericaceae* dwarf shrubs and *Sphagnum fuscum* | 20-30 | 530 |
| 2. | Core 5 Core 19 | Ridge in ridge-hollow complex; covered by low pine (up to 2 m high), *Ericaceae* dwarf shrubs and *Sphagnum fusum* | 15-20 | 390 440 |
| 3. | Core 5-5 | Ecotone between ridge and hollow; covered by mixed species from both habitats: cotton-grass, *Sphagnum* mosses (*S. fusum, S. balticum*), *Ericaceae* dwarf shrubs | 5-10 | 310 |
| 4 | Core 18 | Floating *Sphagnum* mat close at the lake; covered by *Scheuchzeria*, sedges (*Carex limosa*), *Sphagnum* mosses (*S. pappilosum, S. balticum*) | 2-5 | 480 |
| 5 | Core 27 | Ridge in ridge-pool complex; treeless ridge with dwarf shrubs and *Sphagnum* mosses | 10-15 | 400 |
| 6 | Core 31 | Hollow in ridge-hollow complexes; covered by sedges (*Carex limosa*) and *Sphagnum balticum* | 5-10 | 380 |





For the description the peatland stratigraphy additionally 34 peat cores has been collected for the period 2010-2016 (Fig. 1).

**3.2. Plant macrofossils**

Plant macrofossil was analyzed in each 10-cm subsample. For that a piece of ~10 cm$^3$ was sieved through 0.25 mm mesh under flowing water. Plant remains were identified under the binocular microscope (Zeiss Axiostar, 10–40×magnification, Jena, Germany) using both our own experience and the key samples data bank. Peat content was described as abundance of each type of plant remains in percepts, and dominated species remains in a sample determined a peat type.

**3.3. Bulk density, carbon and ash content**

Bulk density, carbon and ash content were determined for each 10 cm subsample using the middle part of 5 cm length (volume 50 cm$^3$). Bulk density (BD: g cm$^{-3}$) was measured by drying the peat at 105° C for 24 hours and later weighting. The dried subsample was grinded and divided into two parts. Ash content was determined from one part by ignition (Nabertherm L9/11/SKM, Lilienthal,

Germany) at 525° C for 9 hours, and a carbon content was determined in the other part. In the elemental analyzer (EA-3000, EuroVector, Pavia, Italy) sample combusted in the oxygen and helium flow on the Ni/Cu catalysts and separates on the chromatography column. Elements are identified on the thermal conductivity detector using Atropine (C=70.56 %, N=4.84 %, H=8.01 %, O=16.59 %) as a standard.

Bulk density and ash content have been defined only for the cores 5a, 5_5, 19a, 27, carbon content – for the cores 2, 5a, 5_5, 19a.

**3.4. Accumulation rates and dissolved organic carbon downward velocity calculations**

Peat accumulation rate (A) was calculated for each 50 cm part (or more in case of a dating lack) using the next Eq. (1):

$$A = (d_l-d_u)/(a_l-a_u) ,\qquad\qquad\qquad (1)$$

where A - peat accumulation rate, cm y$^{-1}$, $d_l$ – lower dated depth, $d_u$ - upper dated depth, $a_l$ – the date of lower depth, $a_u$ - the date of upper depth.

Carbon accumulation rate (CAR) was calculated for each 10 cm part using Eq. (2):

$$CAR = ((V*BD*LOI/100)*CC/100) / (10/A_i) ,\qquad\qquad (2)$$

where CAR - carbon accumulation rate, g C m$^{-2}$ y$^{-1}$, V – volume of peat 10000, cm$^3$ (i.e. 1 m$^2$ of 10 cm depth), BD – bulk density, g cm$^{-3}$, LOI – loss on ignition, %, CC – carbon content, %, 10/A$_i$ – years to grow 10 cm of peat, years.



For carbon accumulation rate calculations for cores 2, 18, 27, 31 were used the mean values of necessary properties based on statistical analysis of all analyzed cores from the Mukhrino mire.

Long term rate of carbon accumulation (LORCA) was using Eq. (3):

$$LORCA = CAR_{cumulative} / A_{bottom} , \qquad\qquad (3)$$

where LORCA - long term rate of carbon accumulation, g C $m^{-2}$ $yr^{-1}$, $CAR_{cumulative}$ - cumulative carbon storage in a square meter in the core, g C $m^{-2}$, $A_{bottom}$ – the bottom age of the core, years. DOC downward movement velocity was calculated  The next Eq. (4) has been used:

$$v = (d_i - d_{i-1})/(a_{poc} - a_{doc}) , \qquad\qquad (4)$$

where $v$ – DOC movement velocity, cm $yr^{-1}$, $d_i$ – depth of current DOC age, cm, $d_{doc\_i}$ – POC depth of the same age as a current DOC, cm, $a_{poc}$ – age of POC on the current depth, years, $a_{doc}$ – age of DOC on the current depth, years.

### 3.5. Separation of DOC and POC

Using the approach from (Schulze, et al., 2015) it is possible to separate the DOC from the POC by dispersing the peat sample in distilled water. First the frozen peat sample was thrown carefully, weighed, dissolved in distilled water and shaked for 2 hours. The solution was wet-sieved with a 63 and 36 μm mesh sieve. The residues from the sieve were freeze dried (Piatkowski, Munich, Germany). The sieved solution (< 36 μm) was adjusted to pH 9 by adding NaOH, shaked another 20

Minutes and centrifuged at 2900 g for 30 min (Megafuge 3.0, Heraeus, Hanau, Germany). The obtained supernatant was filtered with a vacuum flask using a 1.6 μm glass fiber filter (Sartorius) which has been baked at 500° C beforehand. The filtered matter < 1.6 μm was freeze dried and defined as dissolved organic matter (DOC).

The filter residues > 1.6 and the pellet remaining from the centrifugation < 36 μm were merged and

freeze-dried. This fraction (> 1.6, < 36 μm) was defined as particulate organic matter (POC).

### 3.6. AMS 14 C analysis

The peat core DOC and POC samples were analyzed with the Accelerator Mass Spectrometer (AMS) radiocarbon ($^{14}$C) method (Steinhof, 2016, 2017).

For one measurement 0.7 mgC is needed. The Samples pass a chemical preparation whereby the

sample was combusted and the emerged $CO_2$ trapped and catalytically reduced to graphite under presence of $Fe^{2+}$ powder and $H_2$. The resulting graphite was pressed into targets and finally measured in the AMS system. The graphite was ionized in the AMS system (negative charge) and accelerated within an electric field to a final energy of 400 keV. The $^{14}$C isotope ratios was been corrected with the measured 13/12C AMS values (Steinhof, 2017). The radiocarbon dates were





calibrated (Appendix. Table A1) with the IntCal20 (Reimer et al., 2020) and NH1 post-bomb (Hua et al., 2013) atmospheric curves using the package 'clam' (Blaauw, 2020). The age-depth model was developed using the Bayesian-based package 'rbacon' (Blaauw et al., 2020) with 95 % confidence intervals.

## 4. Results

*4.1 Peat stratigraphy profile, ages and accumulation rate*

Average peatland depth based on 35 cores is 340 cm ranging from 85 till 530 cm (Fig. 2. age_depth_cores.png). Two deepest points were found at the northern part of the peatland: the western point is a depression of primary lake (Core 18; 480 cm) partly covered by gyttja (100 cm) and peat (380 cm); the point on the east (Core 2; 530 cm) is associated with an ancient streambed

rush-peat at the bottom. Other cores have minerotrophic grass-woody peat at the bottom (40–60 cm) and have depth till 350-400 cm.

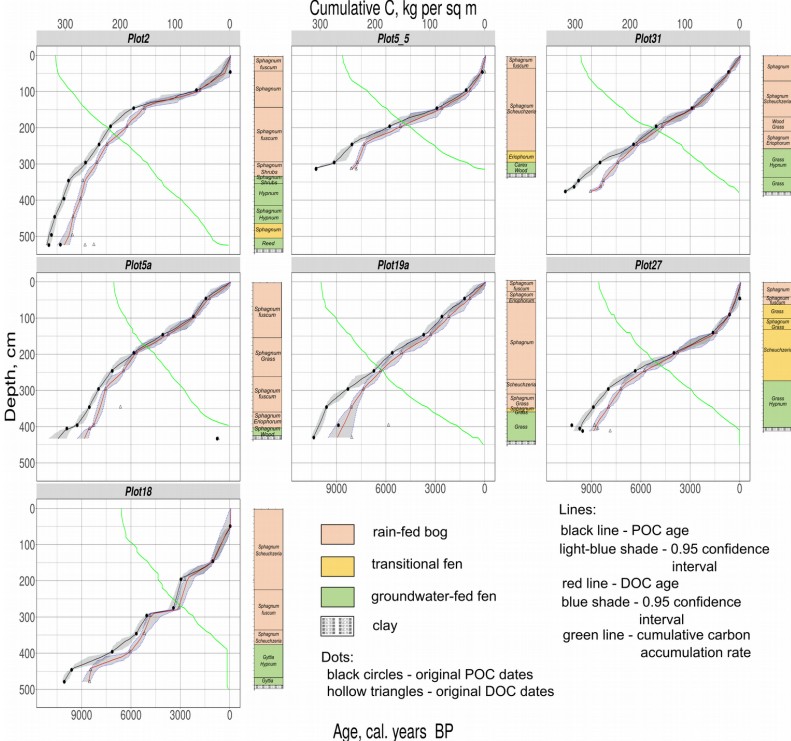

**Figure 2: The age-depth model, carbon accumulation curve and peat types from the Mukhrino peatland.**



Shallow peat deposits (~100 cm and less) are presented on the peatland edge at the border with forest and mineral islands. This zone is affected by frequent fires characterized by abundance of charcoal in peat deposits and admixture of *Carex globularis* in the modern vegetation cover as an indicator of fire disturbance (Lamentowich, 2015). The peatland is still spreading via a paludification process covering the surrounding forested areas on mineral soils. These territories are
characterized by abundance of tall, dry and dead conifer trunks with shallow peat deposits.

The average long-term peat accumulation rate is in range 0.03-0.048 cm yr$^{-1}$ (mean value is 0.041±0.007 cm yr$^{-1}$) dependent on the peat core depth, i.e. in the deepest cores a peat accumulation process is in 1.2–1.6 times faster over 10–11 kyr. The average peat accumulation rate based on 64 dates over different peat layers has higher value 0.067±0.018 cm yr$^{-1}$ (0.013–0.332 cm yr$^{-1}$) mostly
because of increased rates (0.137–0.332 cm yr$^{-1}$) of peat accumulation at the recent time (from 600 BP till now) for several cores.

High values of peat accumulation rate found for the bottom minerotrophic layers of Core 2 (0.086–0.277 cm yr$^{-1}$) which shifted till 0.049 cm yr$^{-1}$ about 9800 cal years BP when vegetation changed from minerotrophic to the transition vegetation type. The highest values are found for the last ~1200
230 year for the cores 5-5 and 19a (0.332 and 0.336 cm yr$^{-1}$ correspondingly) which located on the *Sphagnum fuscum* ridges (water level 30-40 cm) in the ridge-hollow complexes. For the same period increased values also found for the cores 18 and 27 (0.137 and 0.147 cm yr$^{-1}$ correspondingly) with high water level (because of hydrophilic plants presence) and dominance of *Sphagnum papilosum* in moss layer. Generally the peat accumulation rate decreases for the period
6500–3000 cal years BP and increases for the older and younger dates reaching maximum values in the current time.

The oldest dates are found in the cores 2 and 18 (~11000 and 10000 cal yr BP) that means a peat accumulation process started from the eastern and western peatland sides and occupied a central part during next 1500–2500 years. Thus, the lateral rate of peatland distribution might be assessed
between 0.65–1.0 m yr$^{-1}$ at the Preboreal and Boreal periods (11 700–8 200 cal yr BP).

The oldest peat is 10 989 cal yr BP, the oldest gyttja is 10 053 cal yr BP. Peat growth started via terrestrialisation when the lake sediments filled the lake basin between 7 000 and 6 000 cal yr BP. The average age of peatland initiation based on 8 dates is 10 265 cal yr BP (Appendix. Table A1).

**4.2. Bulk density and ash content**

Bulk density (BD) values increase linearly with depth from 0.016 till 0.348 g cm$^{-1}$. Mostly it is caused by changes of peat stratigraphy: oligotrophic *Sphagnum* moss peat types have the lowest BD values because they consist only of the *Sphagnum* remains which mostly resistant to the decomposition processes and thus keep their initial volume (Pologova, Lapshina, 2001). For



Mukhrino peatland many well-preserved *Sphagnum fuscum* peats (decomposition rate 5-10 %) have
250 been found on the depth below 200 cm. Minerotrophic peat contains a lot of woody and sedge
remains which over time lost volume structure and create dense peat layers, also it additionally may
mix with mineral sediments at the bottom (Loisel, 2014).

Ash content changes irregular with the local maximum (5–8%) at the 100 cm depth, with the next
decline (~2.5%) and slow increase till the mineral bottom (5–7%). These variations are related to
255 the plant remain composition of peat – oligotrophic *Sphagnum* peat types have the lowest ash
content while herbs and wood peat have increased concentration of ash (Loisel, 2014).

### 4.3. Peat accumulation rate and carbon content

LORCA ranges between 24.80–28.92 g C m$^{-2}$ yr$^{-1}$ (average value is 26.93±1.76 g C m$^{-2}$ yr$^{-1}$). The
carbon accumulation rate was estimated also on a base of averaging data for each 10 cm layer. It has
260 an average value 38.56±12.21 g C m$^{-2}$ yr$^{-1}$ (28.46–57.91 g C m$^{-2}$ yr$^{-1}$). This value exceeds the
LORCA because of irregularity of a carbon accumulation process.

The average C content for the 10 cm layer is 6.16±1.46 kg C m$^{-2}$. The total amount of carbon stored
in 4.3 m depth (average depth of all cores used for calculation) is 264.9±62.8 kg C m$^{-2}$.

### 4.4. POC & DOC

Relation between the DOC and POC dates is well-correlated (r$^2$=0.98, slope 0.93) till the breaking
point at ~6000 cal year BP (Fig. 2). Older dates are with the less slope (0.7) and more distributed
(r$^2$=0.75). Differences between DOC and POC ages from the same depth are linear (r$^2$ equal 0.55,
slope is 0.14) and range from 80 yr at 50 cm untill 2000–3000 yr at the bottom layers (430–530 cm).
The linear (cores 5, 5-5, 18, 19, 27 and 31) and exponential (core 2) models have been used for
calculations d$_{doc\_i}$ (the depth POC came from). The average velocity of DOC downward movement
is 0.047±0.019 cm yr$^{-1}$ (Fig. 2). The minimal –0.52 cm yr$^{-1}$ and maximal 17.7 cm yr$^{-1}$ values are
found for the upper 50 cm and probably related to the dating uncertainty of modern dates. Excluding
those values the rest of calculated DOC movement rates are in range from –0.24 to 0.97 cm yr$^{-1}$.
Negative values mean an upward DOC movement which found for 10 samples (~15 %).

## 5. Discussion

The main physic-chemical peat properties such as carbon and ash contents, bulk density along the
peat profile has been discovered in the current study. It is shown that ~2/3 of a peat body consists
the *sphagnum* oligotrophic peat with low ash-content and low bulk density what nicely match to the
existing data for the West Siberian lowland (Bleuten and Lapshina, 2001). These properties are



mostly determined by composition of peat plant remains and climatic conditions of the time the peat had been formed and less related to the depth it had been extracted from (Chambers et al., 2010).

The *sphagnum* peat bogs are dominant in the middle taiga zone (Peregon et al., 2008), presented mostly on the watersheds and cover ~28 % of whole zonal area (Terentieva, 2016). Generally, such peatlands have similar development history – initial waterlogging via paludification or

terrestrialisation and forming the eutrophic peat layer, short term stage of mesotrophic peat and an abrupt change to the oligotrophic stage (Bolota Zapadnoi…, 2000; Bolotnie sistemy…, 2001).

High ash content values at the upper peatland layers (50–100 cm) has been found in all peat cores in this and other studies at the Mukhrino peatland (Lamentowicz et al., 2015; Tsyganov, et al., 2021). It might be explained by extremely high flooding 1-2 kiloyears ago with alluvial material

sedimentation or by a probable fire occupied a peatland. But the fire history of Mukhrino FS over the last 1300 years have not discovered any significant amounts of charcoal (Lamentowicz et al., 2015).

The average peat accumulation rate (A) is 0.067±0.02 cm yr$^{-1}$. The lowest average value (0.04±0.02 cm yr$^{-1}$) found for the hollow (core 31) and the highest average value (0.10±0.08 cm yr$^{-1}$) found for

the ryam (core 2). The shape of the age-depth model was close to linear for the cores 19a and 31, s-shape for the cores 5a, 27 and 5-5, convex for the core 2 and broken for the core 18. Any features combining all cores into these groups have not been found. The majority of peat age-depth published models have concave shapes, meaning that a decay process is ongoing in the catotelm - lower anaerobic peat layers (Yu, 2001). Absence of this shape model at the Mukhrino peatland may

be caused by dominance of peat-moss (oligotrophic *Sphagnum*) remains (till 90 % of core) which are the most resistant to decomposition (Thormann et al., 1999).

The A was the highest for oligotrophic peat (0.080±0.038 cm yr$^{-1}$), less for minerotrophic peat (0.062±0.033 cm yr$^{-1}$) and the lowest for transitional peat types (0.061±0.027 cm yr$^{-1}$). Oligotrophic peats mostly consist the remains of *Sphagnum* mosses, which are mostly resistant to the

decomposition process. (Pologova and Lapshina, 2001) showed similar values for Great Vasyugan mire, where oligotrophic *Sphagnum* peat has higher A (0.115 cm yr$^{-1}$) then minerotrophic peat (0.059 cm yr$^{-1}$). Probably, these higher values explained by the location of Great Vasyugan mire in Southern taiga, the most favorable meteorological condition zone for peatlands development (Ivanov and Novikov, 1976), and the local plant biodiversity. In (Lapshina, 2011) the average A for

middle taiga zone is 0.056 cm yr$^{-1}$, when in Southern Taiga zone this value is 0.074–0.08 cm yr$^{-1}$, that supports the concept of the different external conditions of peat accumulation.

In case of eutrophic phase of peatland development, the peat accumulation process is determined by the mineral soil proximity and hence favorable geochemical conditions together with faster peat accumulation (Frolking et al., 2001) due to higher litter input (Thormann et al., 1999). Also the fen



vegetation is less sensitive to climate conditions thus has more stable A (Frolking et al., 2001). Nonetheless, initial mass loss rate for fen plant species and older age, i.e. longer priod of decomposition, causes lower A value.

Altogether this means that the type of water supply (rain or ground water) and hence a way of nutrient income is one of the main limiting factors in peat accumulation process.

The average rate of carbon accumulation (CAR) is $37.99\pm11.4$ gC m$^{-2}$ yr$^{-1}$ (median value is 26.17 gC m$^{-2}$ yr$^{-1}$). This value inconsistency is caused by the skewed data to the high values mostly at the upper layers due to high A (0.15–0.33 cm yr$^{-1}$) and at the bottom layers due to higher bulk density (0.4-1.4 g cm$^{-3}$). This value is similar to the average values for the middle boreal zone of Western Siberia ($24.8\pm5.5$ gC m$^{-2}$ yr$^{-1}$) (Lapshina and Pologova, 2011).

A succession of fen-to-bog passed a transitional phase featured to the smallest A (0.037 cm yr$^{-1}$) and CAR (30.46 gC m$^{-2}$ yr$^{-1}$). Thus, reasons are probably related to the composition of peatland vegetation and its high decomposability caused by the lack of *Sphanga* contributing to the slow CAR (Bellen et al., 2011).

The highest CAR is found for the transitional and eutrophic peats, $76.0\pm65.9$ gC m$^{-2}$ yr$^{-1}$ and

$63.1\pm48.0$ gC m$^{-2}$ yr$^{-1}$ correspondingly. The reason is abundance in peat the grass and woody debris, which are rich of carbon. Oligotrophic peat mostly consists the *Sphagnum* mosses remains which contains the lowest amount of carbon. Thus, the lowest value of CAR ($34.4\pm12.1$ gC m$^{-2}$ yr$^{-1}$) found for oligotrophic peat.

Despite of the different view of the age-depth models all cumulative carbon accumulation curves

have similar slightly positive exponential shape. It looks that A does not strongly affect CAR and does stay under control of other factors as vegetation remains diversity and its biochemical content, bulk density, carbon content and local topography and hydrology (Pologova and Lapshina, 2001; Pologova and Lapshina, 2002; Lapshina and Pologova, 2011).

Our results considered the hypothesis about DOC downward movement. Numerous studies have

analyzed DOC export from peatlands (Freeman et al., 2001; Frey and Smith, 2005; Buzek et. al., 2019; Blueten et al., 2020) but only few are related to the downward DOC movement and none covers this process in peatlands of Western Siberia, except (Schulze at al., 2015). In long-term processes as peatland development spanning last 10–12 kyr (Kremenetski et al., 2003) a DOC downward movement may be considered as an additional and significant path of global carbon cycle

which must be taken into account for calculation of peatland carbon balance.

In our study we have estimated an average value of carbon downward movement as $0.047\pm0.019$ cm yr$^{-1}$. There is a slight tendency to decrease of these values (magnitude 2–10 times) forward to the mineral bottom, probably due to a low vertical hydraulic conductivity for the deep, dense and decomposed peats (Beckwith et al., 2003). Another possible reason is complete saturation of pore



water by DOC which concentration systematically increases with depth and may prevent its further active penetration. Limited number of papers covers this topic (Chasar et al., 2000; Cole et al., 2002, Clymo and Bryant, 2008) and report concentrations ~2 mmol dm$^{-3}$ at the surface and 6–22 mmol dm$^{-3}$ at the bottom. There was not found any information for the West Siberian peatlands.

The negative values of DOC velocity mean an upward flux which might be caused by water table

movement at the surface layers. A rising water table may catch a part of DOC produced in the lower layers and move it up to the surface thereby making converse between DOC and POC ages (Schulze et al., 2015). Several negative DOC velocities were found on the deeper layers (200–300 cm). This caused by the methodological flaw in the value calculations when the s-shape (cores 5a and 27) age-depth model is approximated by the linear regression.

There are few publications estimating DOC vertical velocity values. (Charman et al., 1999) used a vertical hydraulic conductivity equal to 31.5 cm yr$^{-1}$ to estimate DOC vertical transport in the UK. This value exceeds our results in ~600 times because based on a potential water movement which significantly ranges under conditions of saturation and peat physical properties (Chabson and Siegel, 1986). This value might be used as a potential rate of DOC downward movement but has to

be considered as a maximum possible velocity, i.e. a limiting factor.

The difference between DOC and POC ages in our study increases with depth (from 9 till 3 044 yr, excluding three negative differences found for the upper 50 cm) that supported only by the concept of DOC dowanward motion. Another reasons of date differences might be largely excluded:

      - sedge and *Scheuchzeria* roots growing down the peat till the 2 meters (Glaser et al. 2012)

have not been found in any dating sample (because of visual control of the peat samples for dating) and can not penetrate in the deeper layers. These remains would cause extreme inversions of the age-depth model (for example, when the modern roots reach the ancient peat layers) what not found in current study. The roots of trees and dwarf-shrubs occupy only the surface aerobic layer due to absence of aerenchyme cells;

- permafrost has not been distributed in the middle taiga zone last centuries and thus the cryoturbation can not cause so intensive and ubiquitous date discrepancy;

      - periodical flooding of the Mukhrino peatland has to form an alluvium layer which, probably, has been detected once only in the upper layer;

      - peatfires, which happens only in the extrimely hot and dry years in the taiga zone, do not

explain behavior of the DOC and POC ages along a peat profile as well.

So, this process might be caused only by the continuous vertical flow of DOC over the thousands years while the POC stays on the same depth. Thus, the moving DOC penetrates deeper every year and the date discrepancy increases.



This process is facilitated by the local relief position where Mukhrino peatland occupies the high

second terrace and drained by the small rivers "Mukhrina" and "Bolshaya rechka" located ~6-8 m lower from the eastern and western sides. This creates the piezometric gradient and allows the peatland water penetrates through the mineral bottom (clay layer with low hydraulic conductivity) and discharges to the streams (Fig. 3).

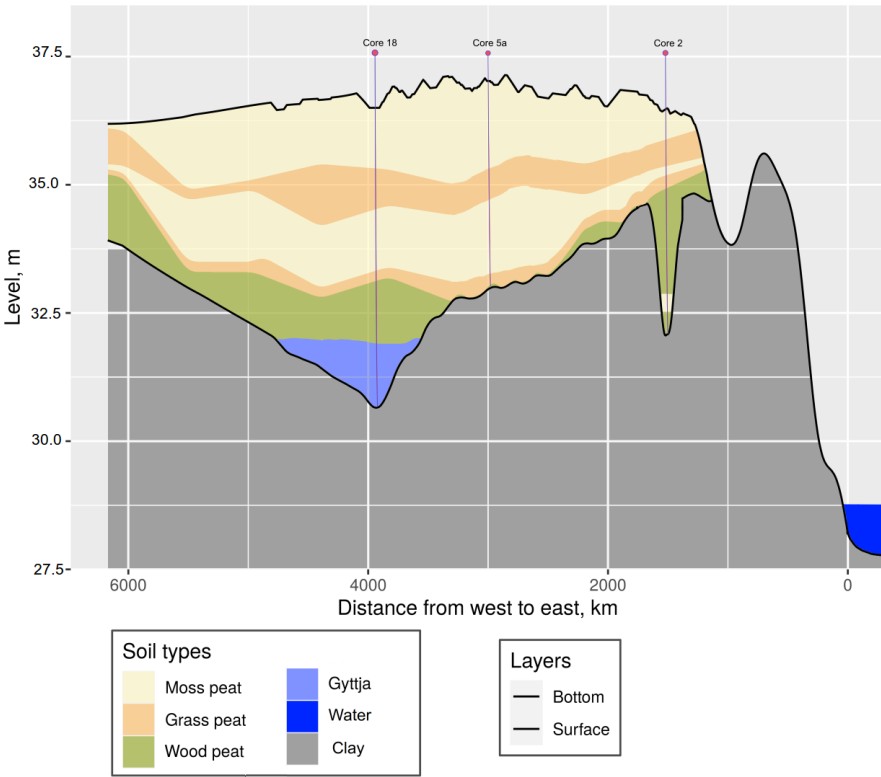

**Figure 3: Peatland stratigraphy and shape of mineral bottom.**

The mechanism of DOC downward movement is described in number of publications (Aravena et al., 1993; Charman et al., 1993, 1994, 1999; Chanton et al., 1995) showing that the gas [14]C ages are

significantly younger in DOC than the particulate peat. The possible reason may be described by conversion of young DOC transported from the upper peat layers to $CO_2$ and $CH_4$ by microbial activity. In our previous study in Western Siberia the maximal age difference 6 500 years has been recorded (Schulze et al., 2015). The results from Southwest England show that DOC is 830–1260yr younger than the peat it was extracted from (Charman et al., 1999). Clymo and Bryant (2008) has





published perfect results about a 7 m deep peatland where differences between the peat and DOC ages increase with depth since 80 till 1 835 yr.

In (Kraev et al., 2017) shown a possible way of methane displacement to the deeper soil horizons due to freezing of thick strata of epigenetic permafrost. The same mechanism potentially might be found for the peatlands, because high peat porosity is a favorable substrate for vertical water

movement. The surface Mukhrino peatland layer freezes at the end of September - beginning of November. The water discharge completely stops at that time, thus a peatland becomes a huge reservoir filled by the high porosity substrate and water, which completely confined by ice pack from above. Frozen water may produce an extra pressure and pushes the labile dissolved form of carbon down.

**Author Contributions:** Conceptualization, E.L. and E.S.; methodology, E.S., I.K., E.Z.; software, E.Z.; investigation, E.Z., I.K.; writing—original draft preparation, E.Z., E.D.; writing—review and editing, E.S., I.K., E.D., E.Z.; visualization, E.Z.; supervision, E.S. and E.L.; All authors have read and agreed to the published version of the manuscript.

**Acknowledgments**: The research was carried out within the grant of the Tyumen region Government in accordance with the program of the West Siberian Interregional Scientific and Educational Centre (National Project "Nauka") and within the grant of the Yugra State University №17-02-07/66.

**Conflicts of Interest:** The authors declare no conflict of interest. The funders had no role in the design of the study; in the collection, analyses, or interpretation of data; in the writing of the manuscript, or in the decision to publish the results.

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





Appendix

Table A1. Radiocarbon dates of the peat core from the Mukhrino peatland.

| Lab number | Plot | Depth, cm | Fraction | Δ14C (‰) | err (‰) | PMC [%] | err PMC [%] | C14 BP | Calibrated age BP (95 % confidence interval) | Calibrated age BP error |
|---|---|---|---|---|---|---|---|---|---|---|
| 18831 | Plot18 | 49 | <1,6 μm | 148.4 | 1.7 | 115.79 | 0.17 | -1177,7 | -24.79 | 11,8 |
| 18830 | Plot18 | 49 | <36>1,6 μm | 107.1 | 1.8 | 111.62 | 0.18 | -883,1 | -27.325 | 12,9 |
| 18822 | Plot18 | 146 | <36>1,6 μm | -138.2 | 1.4 | 86.89 | 0.14 | 1128,9 | 1018,0 | 12,9 |
| 18823 | Plot18 | 146 | <1,6 μm | -139 | 1.9 | 86.81 | 0.19 | 1136,3 | 1067.5 | 17,6 |
| 18824 | Plot18 | 196 | <36>1,6 μm | -304.3 | 1.2 | 70.14 | 0.12 | 2849,1 | 2965.5 | 13,7 |
| 18825 | Plot18 | 196 | <1,6 μm | -279.9 | 1.3 | 72.6 | 0.13 | 2572,2 | 2732.5 | 14,4 |
| 18999 | Plot18 | 275 | <36>1,6 μm | -334.5 | 1.1 | 67.1 | 0.11 | 3205,1 | 3419.5 | 13,2 |
| 19000 | Plot18 | 275 | <1,6 μm | -311 | 1.2 | 69.46 | 0.12 | 2927,4 | 3079,0 | 13,9 |
| 19001 | Plot18 | 296 | <36>1,6 μm | -428.2 | 1 | 57.66 | 0.1 | 4423,0 | 5046.5 | 13,9 |
| 19002 | Plot18 | 296 | <1,6 μm | -416 | 1.1 | 58.88 | 0.11 | 4254,8 | 4841,0 | 15,0 |
| 18826 | Plot18 | 346 | <36>1,6 μm | -466.2 | 1.2 | 53.82 | 0.12 | 4976,6 | 5674,0 | 17,9 |
| 18827 | Plot18 | 346 | <1,6 μm | -436.6 | 1.4 | 56.81 | 0.14 | 4542,3 | 5184,0 | 19,8 |
| 19004 | Plot18 | 396 | <1,6 μm | -484.2 | 1 | 52 | 0.1 | 5253,0 | 6053,0 | 15,4 |
| 19003 | Plot18 | 396 | <36>1,6 μm | -544 | 0.9 | 45.97 | 0.09 | 6243,1 | 7140.5 | 15,7 |
| 18821 | Plot18 | 446 | <1,6 μm | -618 | 0.9 | 38.52 | 0.09 | 7663,4 | 8458,0 | 18,7 |
| 18820 | Plot18 | 446 | <36>1,6 μm | -662 | 0.8 | 34.08 | 0.08 | 8647,2 | 9609,0 | 18,8 |
| 18828 | Plot18 | 479 | <36>1,6 μm | -673.7 | 0.8 | 32.9 | 0.08 | 8930,3 | 10052.5 | 19,5 |
| 18829 | Plot18 | 479 | <1,6 μm | -621.9 | 1.1 | 38.12 | 0.11 | 7747,3 | 8520,0 | 23,1 |
| 18894 | Plot19a | 46 | <36>1,6 μm | -152.9 | 1.5 | 85.4 | 0.15 | 1267,8 | 1224.5 | 14,1 |
| 18895 | Plot19a | 46 | <1,6 μm | -124.5 | 1.6 | 88.27 | 0.16 | 1002,3 | 894.5 | 14,5 |
| 18896 | Plot19a | 96 | <36>1,6 μm | -272 | 1.4 | 73.4 | 0.14 | 2484,2 | 2601,0 | 15,3 |
| 18897 | Plot19a | 96 | <1,6 μm | -241.9 | 1.5 | 76.44 | 0.15 | 2158,2 | 2180,0 | 15,7 |
| 18898 | Plot19a | 146 | <36>1,6 μm | -353 | 1.2 | 65.23 | 0.12 | 3432,1 | 3706.5 | 14,8 |
| 18899 | Plot19a | 146 | <1,6 μm | -322.3 | 1.4 | 68.33 | 0.14 | 3059,1 | 3279.5 | 16,4 |
| 18901 | Plot19a | 196 | <1,6 μm | -427.9 | 1.3 | 57.68 | 0.13 | 4420,2 | 5053,0 | 18,1 |
| 18900 | Plot19a | 196 | <36>1,6 μm | -459.3 | 1.3 | 54.51 | 0.13 | 4874,3 | 5617.5 | 19,1 |
| 18903 | Plot19a | 246 | <1,6 μm | -500.4 | 1.1 | 50.38 | 0.11 | 5507,2 | 6334,0 | 17,5 |
| 18902 | Plot19a | 246 | <36>1,6 μm | -525.6 | 1.1 | 47.83 | 0.11 | 5924,5 | 6732,0 | 18,5 |
| 18905 | Plot19a | 296 | <1,6 μm | -556.6 | 1 | 44.7 | 0.1 | 6468,1 | 7377,0 | 18,0 |
| 18904 | Plot19a | 296 | <36>1,6 μm | -611.8 | 1 | 39.14 | 0.1 | 7535,2 | 8309.5 | 20,5 |
| 18907 | Plot19a | 346 | <1,6 μm | -601.8 | 1 | 40.15 | 0.1 | 7330,5 | 8108.5 | 20,0 |
| 18906 | Plot19a | 346 | <36>1,6 μm | -663 | 1 | 33.98 | 0.1 | 8670,8 | 9613.5 | 23,6 |
| 18909 | Plot19a | 396 | <1,6 μm | -476.3 | 1.2 | 52.8 | 0.12 | 5130,3 | 5845.5 | 18,2 |
| 18908 | Plot19a | 396 | <36>1,6 μm | -634.2 | 1 | 36.88 | 0.1 | 8012,9 | 8889,0 | 21,8 |
| 18911 | Plot19a | 430 | <1,6 μm | -597.8 | 1 | 40.56 | 0.1 | 7248,9 | 8076.5 | 19,8 |
| 18910 | Plot19a | 430 | <36>1,6 μm | -685.5 | 0.9 | 31.71 | 0.09 | 9226,2 | 10386.5 | 22,8 |
| 18815 | Plot2 | 46 | <1,6 μm | 24.6 | 1.6 | 103.31 | 0.16 | -261,6 | -6.5 | 12,4 |
| 18814 | Plot2 | 46 | <36>1,6 μm | 61.5 | 1.8 | 107.03 | 0.18 | -545,8 | -33.185 | 13,5 |
| 18801 | Plot2 | 96 | <1,6 μm | -217.4 | 1.2 | 78.91 | 0.12 | 1902,7 | 1805,0 | 12,2 |
| 18800 | Plot2 | 96 | <36>1,6 μm | -232.7 | 1.3 | 77.36 | 0.13 | 2062,1 | 2022,0 | 13,5 |
| 18796 | Plot2 | 146 | <36>1,6 μm | -475.1 | 1.1 | 52.93 | 0.11 | 5110,6 | 5837,0 | 16,7 |
| 18797 | Plot2 | 146 | <1,6 μm | -437.9 | 1.2 | 56.67 | 0.12 | 4562,1 | 5195,0 | 17,0 |
| 18809 | Plot2 | 196 | <1,6 μm | -496.7 | 1 | 50.75 | 0.1 | 5448,5 | 6249.5 | 15,8 |
| 18808 | Plot2 | 196 | <36>1,6 μm | -549.4 | 1 | 45.43 | 0.1 | 6338,0 | 7242,0 | 17,7 |
| 18803 | Plot2 | 246 | <1,6 μm | -560.7 | 0.9 | 44.29 | 0.09 | 6542,2 | 7459.5 | 16,3 |



| 18802 | Plot2 | 246 | <36>1,6 µm | -591.4 | 0.9 | 41.2 | 0.09 | 7123,1 | 7940,0 | 17,5 |
| 18798 | Plot2 | 296 | <36>1,6 µm | -629 | 0.8 | 37.4 | 0.08 | 7900,5 | 8763.5 | 17,2 |
| 18799 | Plot2 | 296 | <1,6 µm | -597.5 | 0.9 | 40.58 | 0.09 | 7244,9 | 8073.5 | 17,8 |
| 18810 | Plot2 | 346 | <36>1,6 µm | -667.9 | 0.7 | 33.49 | 0.07 | 8787,5 | 9797,0 | 16,8 |
| 18811 | Plot2 | 346 | <1,6 µm | -637 | 1 | 36.6 | 0.1 | 8074,1 | 8938,0 | 21,9 |
| 18817 | Plot2 | 396 | <1,6 µm | -638.6 | 1 | 36.44 | 0.1 | 8109,3 | 9058.5 | 22,0 |
| 18816 | Plot2 | 396 | <36>1,6 µm | -674.9 | 1.7 | 32.78 | 0.17 | 8959,6 | 10070.5 | 41,6 |
| 18813 | Plot2 | 446 | <1,6 µm | -658.6 | 0.8 | 34.42 | 0.08 | 8567,5 | 9521.5 | 18,6 |
| 18812 | Plot2 | 446 | <36>1,6 µm | -692.7 | 0.8 | 30.98 | 0.08 | 9413,3 | 10640,0 | 20,7 |
| 18807 | Plot2 | 496 | <1,6 µm | -660.1 | 0.8 | 34.27 | 0.08 | 8602,5 | 9563.5 | 18,7 |
| 18806 | Plot2 | 496 | <36>1,6 µm | -695.7 | 0.8 | 30.68 | 0.08 | 9491,5 | 10833,0 | 20,9 |
| 18805 | Plot2 | 523 | <1,6 µm | -606.8 | 0.8 | 39.64 | 0.08 | 7433,2 | 8257.5 | 16,2 |
| 18804 | Plot2 | 523 | <36>1,6 µm | -680.9 | 0.9 | 32.17 | 0.09 | 9110,5 | 10289.5 | 22,4 |
| 18818 | Plot2 | 524 | <36>1,6 µm | -701.1 | 0.8 | 30.13 | 0.08 | 9636,8 | 10989,0 | 21,3 |
| 18819 | Plot2 | 524 | <1,6 µm | -630.1 | 1 | 37.3 | 0.1 | 7922,0 | 8791.5 | 21,5 |
| 18873 | Plot27 | 46 | <1,6 µm | 38.6 | 1.6 | 104.72 | 0.16 | -370,5 | -33.09 | 12,3 |
| 18872 | Plot27 | 46 | <36>1,6 µm | 16.3 | 1.8 | 102.47 | 0.18 | -196,0 | -6.105 | 14,1 |
| 18874 | Plot27 | 90 | <36>1,6 µm | -86.6 | 1.5 | 92.1 | 0.15 | 661,1 | 614,0 | 13,1 |
| 18875 | Plot27 | 90 | <1,6 µm | -83.8 | 1.6 | 92.38 | 0.16 | 636,7 | 605,0 | 13,9 |
| 18876 | Plot27 | 140 | <36>1,6 µm | -197.9 | 1.3 | 80.88 | 0.13 | 1704,6 | 1617,0 | 12,9 |
| 18877 | Plot27 | 140 | <1,6 µm | -180.6 | 1.4 | 82.62 | 0.14 | 1533,6 | 1413.5 | 13,6 |
| 18879 | Plot27 | 196 | <1,6 µm | -359.6 | 1.2 | 64.57 | 0.12 | 3513,8 | 3772,0 | 14,9 |
| 18878 | Plot27 | 196 | <36>1,6 µm | -370.6 | 1.2 | 63.46 | 0.12 | 3653,1 | 3990.5 | 15,2 |
| 18880 | Plot27 | 246 | <36>1,6 µm | -503.2 | 1 | 50.09 | 0.1 | 5553,6 | 6346,0 | 16,0 |
| 18881 | Plot27 | 246 | <1,6 µm | -468.5 | 1.1 | 53.59 | 0.11 | 5011,0 | 5772,0 | 16,5 |
| 18883 | Plot27 | 296 | <1,6 µm | -547.4 | 1.2 | 45.64 | 0.12 | 6301,0 | 7214.5 | 21,1 |
| 18882 | Plot27 | 296 | <36>1,6 µm | -594.3 | 1.1 | 40.91 | 0.11 | 7179,9 | 7983.5 | 21,6 |
| 18885 | Plot27 | 346 | <1,6 µm | -592.8 | 1 | 41.06 | 0.1 | 7150,5 | 7975,0 | 19,5 |
| 18884 | Plot27 | 346 | <36>1,6 µm | -634.3 | 1 | 36.87 | 0.1 | 8015,1 | 8889.5 | 21,8 |
| 18887 | Plot27 | 396 | <1,6 µm | -631.6 | 0.9 | 37.14 | 0.09 | 7956,5 | 8817,0 | 19,4 |
| 18886 | Plot27 | 396 | <36>1,6 µm | -676.6 | 1 | 32.6 | 0.1 | 9003,9 | 10199,0 | 24,6 |
| 18888 | Plot27 | 405 | <36>1,6 µm | -665.6 | 0.8 | 33.72 | 0.08 | 8732,5 | 9718.5 | 19,0 |
| 18889 | Plot27 | 405 | <1,6 µm | -626.3 | 1 | 37.68 | 0.1 | 7840,5 | 8620,0 | 21,3 |
| 18891 | Plot27 | 412 | <1,6 µm | -587.6 | 0.9 | 41.58 | 0.09 | 7049,4 | 7869,0 | 17,4 |
| 18890 | Plot27 | 412 | <36>1,6 µm | -659.3 | 0.8 | 34.35 | 0.08 | 8583,8 | 9540,0 | 18,7 |
| 18854 | Plot31 | 46 | <36>1,6 µm | -96.2 | 1.4 | 91.13 | 0.14 | 746,1 | 676,0 | 12,3 |
| 18855 | Plot31 | 46 | <1,6 µm | -80.8 | 1.5 | 92.68 | 0.15 | 610,6 | 599.5 | 13,0 |
| 18857 | Plot31 | 96 | <1,6 µm | -193.7 | 1.4 | 81.3 | 0.14 | 1663,0 | 1605,0 | 13,8 |
| 18856 | Plot31 | 96 | <36>1,6 µm | -208 | 1.4 | 79.86 | 0.14 | 1806,6 | 1684.5 | 14,1 |
| 18858 | Plot31 | 146 | <36>1,6 µm | -299.3 | 1.3 | 70.64 | 0.13 | 2792,1 | 2901.5 | 14,8 |
| 18859 | Plot31 | 146 | <1,6 µm | -291.2 | 1.6 | 71.47 | 0.16 | 2698,2 | 2803,0 | 18,0 |
| 18861 | Plot31 | 196 | <1,6 µm | -408.6 | 1.1 | 59.63 | 0.11 | 4153,2 | 4702,0 | 14,8 |
| 18860 | Plot31 | 196 | <36>1,6 µm | -428.6 | 1.2 | 57.61 | 0.12 | 4430,0 | 5072.5 | 16,7 |
| 18863 | Plot31 | 246 | <1,6 µm | -494.6 | 1.1 | 50.95 | 0.11 | 5416,9 | 6240,0 | 17,3 |
| 18862 | Plot31 | 246 | <36>1,6 µm | -510.4 | 1.1 | 49.37 | 0.11 | 5669,9 | 6446.5 | 17,9 |
| 18865 | Plot31 | 296 | <1,6 µm | -558.9 | 1.1 | 44.47 | 0.11 | 6509,6 | 7403,0 | 19,8 |
| 18864 | Plot31 | 296 | <36>1,6 µm | -620.6 | 1 | 38.26 | 0.1 | 7717,8 | 8484,0 | 21,0 |
| 18867 | Plot31 | 346 | <1,6 µm | -610.4 | 0.9 | 39.28 | 0.09 | 7506,5 | 8296.5 | 18,4 |
| 18866 | Plot31 | 346 | <36>1,6 µm | -667.4 | 0.9 | 33.54 | 0.09 | 8775,5 | 9789.5 | 21,5 |
| 18869 | Plot31 | 363 | <1,6 µm | -617 | 0.9 | 38.62 | 0.09 | 7642,6 | 8448.5 | 18,7 |
| 18868 | Plot31 | 363 | <36>1,6 µm | -674.4 | 0.8 | 32.83 | 0.08 | 8947,4 | 10059.5 | 19,6 |
| 18871 | Plot31 | 376 | <1,6 µm | -639.2 | 1 | 36.38 | 0.1 | 8122,6 | 9062,0 | 22,1 |
| 18870 | Plot31 | 376 | <36>1,6 µm | -691 | 0.9 | 31.16 | 0.09 | 9366,8 | 10591.5 | 23,2 |
| 19006 | Plot5_5 | 46 | <1,6 µm | 3.6 | 1.3 | 101.18 | 0.13 | -94,2 | -5.82 | 10,3 |





| 19005 | Plot5_5 | 46 | <36>1,6 μm | -20.7 | 1.4 | 98.73 | 0.14 | 102,7 | 144.5 | 11,4 |
| 19008 | Plot5_5 | 96 | <1,6 μm | -123.4 | 1.2 | 88.39 | 0.12 | 991,4 | 879,0 | 10,9 |
| 19007 | Plot5_5 | 96 | <36>1,6 μm | -147.2 | 1.3 | 85.99 | 0.13 | 1212,5 | 1123.5 | 12,1 |
| 19010 | Plot5_5 | 146 | <1,6 μm | -274 | 1.1 | 73.2 | 0.11 | 2506,1 | 2608.5 | 12,1 |
| 19009 | Plot5_5 | 146 | <36>1,6 μm | -300 | 1.1 | 70.58 | 0.11 | 2798,9 | 2904.5 | 12,5 |
| 19012 | Plot5_5 | 196 | <1,6 μm | -429.5 | 1 | 57.52 | 0.1 | 4442,6 | 5119,0 | 14,0 |
| 19011 | Plot5_5 | 196 | <36>1,6 μm | -468.6 | 1 | 53.58 | 0.1 | 5012,5 | 5772.5 | 15,0 |
| 19013 | Plot5_5 | 246 | <36>1,6 μm | -597 | 0.8 | 40.63 | 0.08 | 7235,0 | 8070.0 | 15,8 |
| 19014 | Plot5_5 | 246 | <1,6 μm | -552.2 | 0.9 | 45.15 | 0.09 | 6387,7 | 7339.5 | 16,0 |
| 19015 | Plot5_5 | 296 | <36>1,6 μm | -642.1 | 0.9 | 36.09 | 0.09 | 8186,9 | 9145,0 | 20,0 |
| 19016 | Plot5_5 | 296 | <1,6 μm | -579.9 | 1.1 | 42.36 | 0.11 | 6900,1 | 7729.5 | 20,8 |
| 18850 | Plot5_5 | 313 | <36>1,6 μm | -680.4 | 0.8 | 32.22 | 0.08 | 9098,0 | 10243.5 | 19,9 |
| 18851 | Plot5_5 | 313 | <1,6 μm | -600.9 | 1 | 40.24 | 0.1 | 7312,5 | 8105,0 | 19,9 |
| 18853 | Plot5_5 | 314 | <1,6 μm | -583.3 | 1 | 42.01 | 0.1 | 6966,7 | 7809.5 | 19,1 |
| 18852 | Plot5_5 | 314 | <36>1,6 μm | -680.5 | 0.8 | 32.21 | 0.08 | 9100,5 | 10247.5 | 19,9 |
| 18915 | Plot5a | 46 | <1,6 μm | -158.2 | 1.5 | 84.88 | 0.15 | 1316,9 | 1234,0 | 14,2 |
| 18914 | Plot5a | 46 | <36>1,6 μm | -183.4 | 1.7 | 82.34 | 0.17 | 1560,9 | 1451,0 | 16,6 |
| 18917 | Plot5a | 96 | <1,6 μm | -240 | 1.5 | 76.63 | 0.15 | 2138,2 | 2149.5 | 15,7 |
| 18916 | Plot5a | 96 | <36>1,6 μm | -243.9 | 1.5 | 76.24 | 0.15 | 2179,2 | 2211,0 | 15,8 |
| 18919 | Plot5a | 146 | <1,6 μm | -356.3 | 1.4 | 64.9 | 0.14 | 3472,8 | 3738.5 | 17,3 |
| 18918 | Plot5a | 146 | <36>1,6 μm | -376.5 | 1.4 | 62.87 | 0.14 | 3728,1 | 4068,0 | 17,9 |
| 18921 | Plot5a | 196 | <1,6 μm | -465.5 | 1.1 | 53.89 | 0.11 | 4966,2 | 5669,0 | 16,4 |
| 18920 | Plot5a | 196 | <36>1,6 μm | -473.8 | 1.1 | 53.06 | 0.11 | 5090,9 | 5830,0 | 16,6 |
| 18922 | Plot5a | 246 | <36>1,6 μm | -544.2 | 0.9 | 45.96 | 0.09 | 6244,8 | 7141.5 | 15,7 |
| 18923 | Plot5a | 246 | <1,6 μm | -511 | 1 | 49.31 | 0.1 | 5679,7 | 6449,0 | 16,3 |
| 18925 | Plot5a | 296 | <1,6 μm | -569.2 | 0.9 | 43.43 | 0.09 | 6699,7 | 7560,0 | 16,6 |
| 18924 | Plot5a | 296 | <36>1,6 μm | -592.6 | 0.9 | 41.07 | 0.09 | 7148,5 | 7974.5 | 17,6 |
| 18927 | Plot5a | 346 | <1,6 μm | -519.4 | 1 | 48.45 | 0.1 | 5821,0 | 6639.5 | 16,6 |
| 18926 | Plot5a | 346 | <36>1,6 μm | -623 | 0.8 | 38.01 | 0.08 | 7770,5 | 8529,0 | 16,9 |
| 18929 | Plot5a | 396 | <1,6 μm | -606.4 | 0.8 | 39.69 | 0.08 | 7423,1 | 8254,0 | 16,2 |
| 18928 | Plot5a | 396 | <36>1,6 μm | -645.7 | 0.9 | 35.73 | 0.09 | 8267,4 | 9268.5 | 20,2 |
| 18931 | Plot5a | 405 | <1,6 μm | -623.3 | 0.8 | 37.98 | 0.08 | 7776,8 | 8535.5 | 16,9 |
| 18930 | Plot5a | 405 | <36>1,6 μm | -668.3 | 0.8 | 33.45 | 0.08 | 8797,1 | 9898.5 | 19,2 |
| 18912 | Plot5a | 433 | <36>1,6 μm | -110.4 | 1.5 | 89.69 | 0.15 | 874,1 | 759.5 | 13,4 |
| 18913 | Plot5a | 433 | <1,6 μm | -104.5 | 1.6 | 90.29 | 0.16 | 820,5 | 709,0 | 14,2 |