# Peer review of "Dissolved organic carbon vertical movement and carbon accumulation in West Siberian peatlands"

_Biogeosciences, 2021_

## Referee Comment (RC1)

Major comments:

1. First of all, English language of this paper is inappropriate for scientific publishing and especially in Biogeosciences. Paper should be completely revised to improve grammar and use of terminology. For instance, sentences

In (Kraev et al., 2017) shown a possible way of methane displacement to the deeper soil horizons due to freezing of thick strata of epigenetic permafrost. The same mechanism potentially might be found for the peatlands, because high peat porosity is a favorable substrate for vertical water movement.
should be rearranged like
A possible way of methane migration to the deeper soil horizons due to freezing of thick strata of epigenetic permafrost is demonstrated in (Kraev et al., 2017). The same mechanism might be found for peatlands, because high peat porosity provides intensive vertical water movement.

In the first sentence word order is wrong and passive form is missed. Porosity cannot be a substrate as you stated in the second sentence. Please, also check using of articles throughout the paper.

2. Figures and graphics in the paper must be sufficiently improved. Scales should be added in Figure 1. Figure 2 is a main result of this paper but it is very small, it is hard to read data from it. Cap should be much more informative, I suggest to place lines and dots description in the cap, add letters denoting what ecosystem type is presented by corresponding peat core.

3. Did you compare DOC concentration in a water sample after the extraction with in situ concentration of DOC in wetland pore water? Can extraction provide artificial fractionation? My concern is how we can extrapolate results of this paper to a real nature. Since you have any assumptions they should be stated explicitly and their applicability should be assessed somehow.

4. I think it is not correct to designate velocity calculated via equation 4 as a DOC vertical movement velocity. To calculate real DOC transport one needs to consider DOC diffusion, production and consumption rates in a peat profile. Via equation 4 **effective** (observed) vertical movement velocity is calculated and all mentioned mass balance terms for DOC are integrated in this velocity. This velocity does not correspond to the real process of DOC vertical migration, it explains only resulting value for all DOC-related processes. Thus it is not correct to compare your results with values from (Charman et al., 1999).
If this analysis is correct it should be stated in a paper text to help reader interpret your result in a correct manner.

5. Main focus of the paper is not clear for reader. In introduction section you state hypothesis on DOC-POC age differences. But most of the paper and discussion section is about peat history and stratigraphy. Paper title, Introduction and Results-Discussion should be consistent with

each other. Please, declare that you have several certain goals and show their scientific importance in the Introduction. And consider them correspondingly in sections below.

Minor comments:

l 28 West Siberia is not a wetland, please fix this sentence. Also 50-70% sounds too much, wetland area estimated by Terentieva et al. 2016 is much less (about 30%).

l 108 Terms mesotrophic and oligotrophic are not international. Can you use terms "ombrotrophic" and "minerotrophic" instead? Otherwise these terms need definitions.

l 126 In summer 2016 – what was an interval between sampling and AMS analysis? Can it alter results and what was done to avoid this bias?

l 143 key samples data bank – what data bank, does it available somehow?

l 164 I don't understand why both LOI and CC are used in this formula. As you said CC is determined in a dried sample and NOT in the ash. Please give any reference for this formula.

l 173 How did you calculate cumulative CAR?

l 175 This is a main equation for your paper. Is it presented for the first time? Where does it come from? References and details must be added here.

l 176 You did not use $d_{doc\_i}$ in your equation but mention it in the description of terms, please fix it.

l 349 What does it mean – "complete saturation of pore water by DOC"? Was it described somewhere that there is a complete saturation of water in DOC?

l 375 References are needed.

l 394 Gas $^{14}C$ values – what does it mean, what gas?